# Adaptive Primal-Dual Splitting Methods for Statistical Learning and Image Processing

**Thomas Goldstein**[*]
Department of Computer Science
University of Maryland
College Park, MD

**Min Li**[†]
School of Economics and Management
Southeast University
Nanjing, China

**Xiaoming Yuan**[‡]
Department of Mathematics
Hong Kong Baptist University
Kowloon Tong, Hong Kong

## Abstract

The alternating direction method of multipliers (ADMM) is an important tool for solving complex optimization problems, but it involves minimization sub-steps that are often difficult to solve efficiently. The Primal-Dual Hybrid Gradient (PDHG) method is a powerful alternative that often has simpler sub-steps than ADMM, thus producing lower complexity solvers. Despite the flexibility of this method, PDHG is often impractical because it requires the careful choice of multiple stepsize parameters. There is often no intuitive way to choose these parameters to maximize efficiency, or even achieve convergence. We propose self-adaptive stepsize rules that automatically tune PDHG parameters for optimal convergence. We rigorously analyze our methods, and identify convergence rates. Numerical experiments show that adaptive PDHG has strong advantages over non-adaptive methods in terms of both efficiency and simplicity for the user.

## 1 Introduction

Splitting methods such as ADMM [1, 2, 3] have recently become popular for solving problems in distributed computing, statistical regression, and image processing. ADMM allows complex problems to be broken down into sequences of simpler sub-steps, usually involving large-scale least squares minimizations. However, in many cases these least squares minimizations are difficult to directly compute. In such situations, the Primal-Dual Hybrid Gradient method (PDHG) [4, 5], also called the linearized ADMM [4, 6], enables the solution of complex problems with a simpler sequence of sub-steps that can often be computed in closed form. This flexibility comes at a cost – the PDHG method requires the user to choose multiple stepsize parameters that jointly determine the convergence of the method. Without having extensive analytical knowledge about the problem being solved (such as eigenvalues of linear operators), there is no intuitive way to select stepsize parameters to obtain fast convergence, or even guarantee convergence at all.

In this article we introduce and analyze self-adaptive variants of PDHG – variants that automatically tune stepsize parameters to attain (and guarantee) fast convergence without user input. Applying adaptivity to splitting methods is a difficult problem. It is known that naive adaptive variants of

---

[*]tomg@cs.umd.edu

[†]limin@seu.edu.cn

[‡]xmyuan@hkbu.edu.hk

ADMM are non-convergent, however recent results prove convergence when specific mathematical requirements are enforced on the stepsizes [7]. Despite this progress, the requirements for convergence of adaptive PDHG have been unexplored. This is surprising, given that stepsize selection is a much bigger issue for PDHG than for ADMM because it requires multiple stepsize parameters.

The contributions of this paper are as follows. First, we describe applications of PDHG and its advantages over ADMM. We then introduce a new adaptive variant of PDHG. The new algorithm not only tunes parameters for fast convergence, but contains a line search that guarantees convergence when stepsize restrictions are unknown to the user. We analyze the convergence of adaptive PDHG, and rigorously prove convergence rate guarantees. Finally, we use numerical experiments to show the advantages of adaptivity on both convergence speed and ease of use.

## 2 The Primal-Dual Hybrid Gradient Method

The PDHG scheme has its roots in the Arrow-Hurwicz method, which was studied by Popov [8]. Research in this direction was reinvigorated by the introduction of PDHG, which converges rapidly for a wider range of stepsizes than Arrow-Hurwicz. PDHG was first presented in [9] and analyzed for convergence in [4, 5]. It was later studied extensively for image segmentation [10]. An extensive technical study of the method and its variants is given by He and Yuan [11]. Several extensions of PDHG, including simplified iterations for the case that $f$ or $g$ is differentiable, are presented by Condat [12]. Several authors have also derived PDHG as a preconditioned form of ADMM [4, 6].

PDHG solves saddle-point problems of the form

$$\min_{x \in X} \max_{y \in Y} f(x) + y^T A x - g(y). \tag{1}$$

for convex $f$ and $g$. We will see later that an incredibly wide range of problems can be cast as (1).

The steps of PDHG are given by

$$
\begin{cases}
\hat{x}^{k+1} = x^k - \tau_k A^T y^k & (2) \\
x^{k+1} = \underset{x \in X}{\arg\min} \, f(x) + \dfrac{1}{2\tau_k} \|x - \hat{x}^{k+1}\|^2 & (3) \\
\hat{y}^{k+1} = y^k + \sigma_k A(2x^{k+1} - x^k) & (4) \\
y^{k+1} = \underset{y \in Y}{\arg\min} \, g(y) + \dfrac{1}{2\sigma_k} \|y - \hat{y}^{k+1}\|^2 & (5)
\end{cases}
$$

where $\{\tau_k\}$ and $\{\sigma_k\}$ are stepsize parameters. Steps (2) and (3) of the method update $x$, decreasing the energy (1) by first taking a gradient descent step with respect to the inner product term in (1) and then taking a "backward" or proximal step involving $f$. In steps (4) and (5), the energy (1) is increased by first marching up the gradient of the inner product term with respect to $y$, and then a backward step is taken with respect to $g$.

PDHG has been analyzed in the case of constant stepsizes, $\tau_k = \tau$ and $\sigma_k = \sigma$. In particular, it is known to converge as long as $\sigma\tau < 1/\rho(A^T A)$ [4, 5, 11]. However, PDHG typically does not converge when non-constant stepsizes are used, even in the case that $\sigma_k \tau_k < 1/\rho(A^T A)$ [13]. Furthermore, it is unclear how to select stepsizes when the spectral properties of $A$ are unknown. In this article, we identify the specific stepsize conditions that guarantee convergence in the presence of adaptivity, and propose a backtracking scheme that can be used when the spectral radius of $A$ is unknown.

## 3 Applications

**Linear Inverse Problems**   Many inverse problems and statistical regressions have the form

$$\text{minimize} \quad h(Sx) + f(Ax - b) \tag{6}$$

where $f$ (the data term) is some convex function, $h$ is a (convex) regularizer (such as the $\ell_1$-norm), $A$ and $S$ are linear operators, and $b$ is a vector of data. Recently, the alternating direction method

of multipliers (ADMM) has become a popular method for solving such problems. The ADMM relies on the change of variables $y \leftarrow Sx$, and generates the following sequence of iterates for some stepsize $\tau$

$$\begin{cases} x^{k+1} &= \arg\min_x f(Ax - b) + (Sx - y^k)^T \lambda^k + \frac{\tau}{2}\|Sx - y^k\|^2 \\ y^{k+1} &= \arg\min_y h(y) + (Sx^{k+1} - y)^T \lambda^k + \frac{\tau}{2}\|Sx^{k+1} - y\|^2 \\ \lambda^{k+1} &= \lambda^k + \tau(Sx^{k+1} - y^{k+1}). \end{cases} \tag{7}$$

The $x$-update in (7) requires the solution of a (potentially large) least-square problem involving both $A$ and $S$. Common formulations such as the consensus ADMM [14] solve these large sub-problems with direct matrix factorizations, however this is often impractical when either the data matrices are extremely large or fast transforms (such as FFT, DCT, or Hadamard) cannot be used.

The problem (6) can be put into the form (1) using the *Fenchel conjugate* of the convex function $h$, denoted $h^*$, which satisfies the important identity

$$h(z) = \max_y y^T z - h^*(y)$$

for all $z$ in the domain of $h$. Replacing $h$ in (6) with this expression involving its conjugate yields

$$\min_x \max_y f(Ax - b) + y^T Sx - h^*(y)$$

which is of the form (1). The forward (gradient) steps of PDHG handle the matrix $A$ *explicitly*, allowing linear inverse problems to be solved without any difficult least-squares sub-steps. We will see several examples of this below.

**Scaled Lasso**   The square-root lasso [15] or scaled lasso [16] is a variable selection regression that obtains sparse solutions to systems of linear equations. Scaled lasso has several advantages over classical lasso – it is more robust to noise and it enables setting penalty parameters without cross validation [15, 16]. Given a data matrix $D$ and a vector $b$, the scaled lasso finds a sparse solution to the system $Dx = b$ by solving

$$\min_x \mu\|x\|_1 + \|Dx - b\|_2 \tag{8}$$

for some scaling parameter $\mu$. Note the $\ell_2$ term in (8) is not squared as in classical lasso. If we write

$$\mu\|x\|_1 = \max_{\|y_1\|_\infty \le \mu} y_1^T x, \quad \text{and} \quad \|Dx - b\|_2 = \max_{\|y_2\|_2 \le 1} y_2^T(Dx - b)$$

we can put (8) in the form (1)

$$\min_x \max_{\|y_1\|_\infty \le \mu, \|y_2\|_2 \le 1} y_1^T x + y_2^T(Dx - b). \tag{9}$$

Unlike ADMM, PDHG does not require the solution of least-squares problems involving $D$.

**Total-Variation Minimization**   Total variation [17] is commonly used to solve problems of the form

$$\min_x \mu\|\nabla x\|_1 + \frac{1}{2}\|Ax - f\|_2^2 \tag{10}$$

where $x$ is a 2D array (image), $\nabla$ is the discrete gradient operator, $A$ is a linear operator, and $f$ contains data. If we add a *dual* variable $y$ and write $\mu\|\nabla x\|_1 = \max_{\|y\|_\infty \le \mu} y^T \nabla x$, we obtain

$$\max_{\|y\|_\infty \le \mu} \min_x \frac{1}{2}\|Ax - f\|^2 + y^T \nabla x \tag{11}$$

which is clearly of the form (1).

The PDHG solver using formulation (11) avoids the inversion of the gradient operator that is required by ADMM. This is useful in many applications. For example, in compressive sensing the matrix $A$ may be a sub-sampled orthogonal Hadamard [18], wavelet, or Fourier transform [19, 20]. In this case, the proximal sub-steps of PDHG are solvable in closed form using fast transforms because they do not involve the gradient operator $\nabla$. The sub-steps of ADMM involve both the gradient operator and the matrix $A$ simultaneously, and thus require inner loops with expensive iterative solvers.

# 4 Adaptive Formulation

The convergence of PDHG can be measured by the size of the *residuals*, or gradients of (1) with respect to the primal and dual variables $x$ and $y$. These primal and dual gradients are simply

$$p^{k+1} = \partial f(x^{k+1}) + A^T y^{k+1}, \quad \text{and} \quad d^{k+1} = \partial g(y^{k+1}) + A x^{k+1} \tag{12}$$

where $\partial f$ and $\partial g$ denote the sub-differential of $f$ and $g$. The sub-differential can be directly evaluated from the sequence of PDHG iterates using the optimality condition for (3): $0 \in \partial f(x^{k+1}) + \frac{1}{\tau_k}(x^{k+1} - \hat{x}^{k+1})$. Rearranging this yields $\frac{1}{\tau_k}(\hat{x}^{k+1} - x^{k+1}) \in \partial f(x^{k+1})$. The same method can be applied to (5) to obtain $\partial g(y^{k+1})$. Applying these results to (12) yields the closed form residuals

$$p^{k+1} = \frac{1}{\tau_k}(x^k - x^{k+1}) - A^T(y^k - y^{k+1}), \quad d^{k+1} = \frac{1}{\sigma_k}(y^k - y^{k+1}) - A(x^k - x^{k+1}). \tag{13}$$

When choosing the stepsize for PDHG, there is a tradeoff between the primal and dual residuals. Choosing a large $\tau_k$ and a small $\sigma_k$ drives down the primal residuals at the cost of large dual residuals. Choosing a small $\tau_k$ and large $\sigma_k$ results in small dual residuals but large primal errors. One would like to choose stepsizes so that the larger of $p^{k+1}$ and $d^{k+1}$ is as small as possible. If we assume the residuals on step $k+1$ change monotonically with $\tau_k$, then $\max\{p^{k+1}, d^{k+1}\}$ is minimized when $p^{k+1} = d^{k+1}$. This suggests that we tune $\tau_k$ to "balance" the primal and dual residuals.

To achieve residual balancing, we first select a parameter $\alpha_0 < 1$ that controls the aggressiveness of adaptivity. On each iteration, we check whether the primal residual is at least twice the dual. If so, we increase the primal stepsize to $\tau_{k+1} = \tau_k/(1-\alpha_k)$ and decrease the dual to $\sigma_{k+1} = \sigma_k(1-\alpha_k)$. If the dual residual is at least twice the primal, we do the opposite. When we modify the stepsize, we shrink the adaptivity level to $\alpha_{k+1} = \eta\alpha_k$, for $\eta \in (0, 1)$. We will see in Section 5 that this adaptivity level decay is necessary to guarantee convergence. In our implementation we use $\alpha_0 = \eta = .95$.

In addition to residual balancing, we check the following backtracking condition after each iteration

$$\frac{c}{2\tau_k}\|x^{k+1} - x^k\|^2 - 2(y^{k+1} - y^k)^T A(x^{k+1} - x^k) + \frac{c}{2\sigma_k}\|y^{k+1} - y^k\|^2 > 0 \tag{14}$$

where $c \in (0, 1)$ is a constant (we use $c = 0.9$) is our experiments. If condition (14) fails, then we shrink $\tau_k$ and $\sigma_k$ before the next iteration. We will see in Section 5 that the backtracking condition (14) is sufficient to guarantee convergence. The complete scheme is listed in Algorithm 1.

---

**Algorithm 1** Adaptive PDHG

---

1: Choose $x^0$, $y^0$, large $\tau_0$ and $\sigma_0$, and set $\alpha_0 = \eta = 0.95$.
2: **while** $\|p^k\|, \|d^k\| > tolerance$ **do**
3:      Compute $(x^{k+1}, y^{k+1})$ from $(x^k, y^k)$ using the PDHG updates (2-5)
4:      Check the backtracking condition (14) and if it fails set $\tau_k \leftarrow \tau_k/2$, $\sigma_k \leftarrow \sigma_k/2$
5:      Compute the residuals (13), and use them for the following two adaptive updates
6:      **If** $2\|p^{k+1}\| < \|d^{k+1}\|$, **then** set $\tau_{k+1} = \tau_k(1-\alpha_k)$, $\sigma_{k+1} = \sigma_k/(1-\alpha_k)$, and $\alpha_{k+1} = \alpha_k\eta$
7:      **If** $\|p^{k+1}\| > 2\|d^{k+1}\|$, **then** set $\tau_{k+1} = \tau_k/(1-\alpha_k)$, $\sigma_{k+1} = \sigma_k(1-\alpha_k)$, and $\alpha_{k+1} = \alpha_k\eta$
8:      **If** no adaptive updates were triggered, **then** $\tau_{k+1} = \tau_k$, $\sigma_{k+1} = \sigma_k$, and $\alpha_{k+1} = \alpha_k$
9: **end while**

---

# 5 Convergence Theory

In this section, we analyze Algorithm 1 and its rate of convergence. In our analysis, we consider adaptive variants of PDHG that satisfy the following assumptions. We will see later that these assumptions guarantee convergence of PDHG with rate $O(1/k)$.

Algorithm 1 trivially satisfies Assumption **A**. The sequence $\{\phi_k\}$ measures the adaptive aggressiveness on iteration $k$, and serves the same role as $\alpha_k$ in Algorithm 1. The geometric decay of $\alpha_k$ ensures that Assumption **B** holds. The backtracking rule explicitly guarantees Assumption **C**.

---

**Assumptions for Adaptive PDHG**

**A** The sequences $\{\tau_k\}$ and $\{\sigma_k\}$ are positive and bounded.

**B** The sequence $\{\phi_k\}$ is summable, where $\phi_k = \max\left\{\frac{\tau_k - \tau_{k+1}}{\tau_k}, \frac{\sigma_k - \sigma_{k+1}}{\sigma_k}, 0\right\}$.

**C** Either $X$ or $Y$ is bounded, and there is a constant $c \in (0,1)$ such that for all $k > 0$

$$\frac{c}{2\tau_k}\|x^{k+1} - x^k\|^2 - 2(y^{k+1} - y^k)^T A(x^{k+1} - x^k) + \frac{c}{2\sigma_k}\|y^{k+1} - y^k\|^2 > 0.$$

---

## 5.1 Variational Inequality Formulation

For notational simplicity, we define the composite vector $u^k = (x^k, y^k)$ and the matrices

$$M_k = \begin{pmatrix} \tau_k^{-1} I & -A^T \\ -A & \sigma_k^{-1} I \end{pmatrix}, \quad H_k = \begin{pmatrix} \tau_k^{-1} I & 0 \\ 0 & \sigma_k^{-1} I \end{pmatrix}, \quad \text{and} \quad Q(u) = \begin{pmatrix} A^T y \\ -Ax \end{pmatrix}. \qquad (15)$$

This notation allows us to formulate the optimality conditions for (1) as a variational inequality (VI). If $u^\star = (x^\star, y^\star)$ is a solution to (1), then $x^\star$ is a minimizer of (1). More formally,

$$f(x) - f(x^\star) + (x - x^\star)^T A^T y^\star \geq 0 \quad \forall x \in X. \qquad (16)$$

Likewise, (1) is maximized by $y^\star$, and so

$$-g(y) + g(y^\star) + (y - y^\star)^T Ax^\star \leq 0 \quad \forall y \in Y. \qquad (17)$$

Subtracting (17) from (16) and letting $h(u) = f(x) + g(y)$ yields the VI formulation

$$h(u) - h(u^\star) + (u - u^\star)^T Q(u^\star) \geq 0 \quad \forall u \in \Omega, \qquad (18)$$

where $\Omega = X \times Y$. We say $\tilde{u}$ is an approximate solution to (1) with VI accuracy $\epsilon$ if

$$h(u) - h(\tilde{u}) + (u - \tilde{u})^T Q(\tilde{u}) \geq -\epsilon \quad \forall u \in B_1(\tilde{u}) \cap \Omega, \qquad (19)$$

where $B_1(\tilde{u})$ is a unit ball centered at $\tilde{u}$. In Theorem 1, we prove $O(1/k)$ ergodic convergence of adaptive PDHG using the VI notion of convergence.

## 5.2 Preliminary Results

We now prove several results about the PDHG iterates that are needed to obtain a convergence rate.

**Lemma 1.** *The iterates generated by PDHG (2-5) satisfy*

$$\|u^k - u^\star\|_{M_k}^2 \geq \|u^{k+1} - u^k\|_{M_k}^2 + \|u^{k+1} - u^\star\|_{M_k}^2.$$

The proof of this lemma follows standard techniques, and is presented in the supplementary material. This next lemma bounds iterates generated by PDHG.

**Lemma 2.** *Suppose the stepsizes for PDHG satisfy Assumptions A, B and C. Then*

$$\|u^k - u^\star\|_{H_k}^2 \leq C_U$$

*for some upper bound $C_U > 0$.*

The proof of this lemma is given in the supplementary material.

**Lemma 3.** *Under Assumptions A, B, and C, we have*

$$\sum_{k=1}^n \left(\|u^k - u\|_{M_k}^2 - \|u^k - u\|_{M_{k-1}}^2\right) \leq 2C_\phi C_U + 2C_\phi C_H \|u - u^\star\|^2$$

*where $C_\phi = \sum_{k=0}^\infty \phi_k$ and $C_H$ is a constant such that $\|u - u^\star\|_{H_k}^2 \leq C_H \|u - u^\star\|^2$.*

*Proof.* Using the definition of $M_k$ we obtain

$$\sum_{k=1}^{n} \left( \|u^k - u\|_{M_k}^2 - \|u^k - u\|_{M_{k-1}}^2 \right)$$

$$= \sum_{k=1}^{n} \left[ (\frac{1}{\tau_k} - \frac{1}{\tau_{k-1}})\|x^k - x\|^2 + (\frac{1}{\sigma_k} - \frac{1}{\sigma_{k-1}})\|y^k - y\|^2 \right]$$

$$\leq \sum_{k=1}^{n} \phi_{k-1} \left( \frac{1}{\tau_k}\|x^k - x\|^2 + \frac{1}{\sigma_k}\|y^k - y\|^2 \right)$$

$$= \sum_{k=1}^{n} \phi_{k-1}\|u^k - u\|_{H_k}^2 \qquad (20)$$

$$\leq 2 \sum_{k=1}^{n} \phi_{k-1} \left( \|u^k - u^\star\|_{H_k}^2 + \|u - u^\star\|_{H_k}^2 \right)$$

$$\leq 2 \sum_{k=1}^{n} \phi_{k-1} \left( C_U + C_H\|u - u^\star\|^2 \right)$$

$$\leq 2C_\phi C_U + 2C_\phi C_H\|u - u^\star\|^2,$$

where we have used the bound $\|u^k - u^\star\|_{H_k}^2 \leq C_U$ from Lemma 2 and $C_\phi = \sum_{k=0}^{\infty} \phi_k$.

$\square$

This final lemma provides a VI interpretation of the PDHG iteration.

**Lemma 4.** *The iterates $u^k = (x^k, y^k)$ generated by PDHG satisfy*

$$h(u) - h(u^{k+1}) + (u - u^{k+1})^T[Qu^{k+1} + M_k(u^{k+1} - u^k)] \geq 0 \qquad \forall u \in \Omega. \qquad (21)$$

*Proof.* Let $u^k = (x^k, y^k)$ be a pair of PDHG iterates. The minimizers in (3) and (5) of PDHG satisfy the following for all $x \in X$

$$f(x) - f(x^{k+1}) + (x - x^{k+1})^T[A^T y^{k+1} - A^T(y^{k+1} - y^k) + \frac{1}{\tau_k}(x^{k+1} - x^k)] \geq 0, \qquad (22)$$

and also for all $y \in Y$

$$g(y) - g(y^{k+1}) + (y - y^{k+1})^T[-Ax^{k+1} - A(x^{k+1} - x^k) + \frac{1}{\sigma_k}(y^{k+1} - y^k)] \geq 0. \qquad (23)$$

Adding these two inequalities and using the notation (15) yields the result. $\square$

### 5.3 Convergence Rate

We now combine the above lemmas into our final convergence result.

**Theorem 1.** *Suppose that the stepsizes in PDHG satisfy Assumptions **A**, **B**, and **C**. Consider the sequence defined by*

$$\tilde{u}_t = \frac{1}{t} \sum_{k=1}^{t} u^k.$$

*This sequence satisfies the convergence bound*

$$h(u) - h(\tilde{u}_t) + (u - \tilde{u}_t)^T Q(\tilde{u}_t) \geq \frac{\|u - \tilde{u}_t\|_{M_t}^2 - \|u - u^0\|_{M_0}^2 - 2C_\phi C_U - 2C_\phi C_H\|u - u^\star\|^2}{2t}.$$

*Thus $\tilde{u}_t$ converges to a solution of (1) with rate $O(1/k)$ in the VI sense (19).*

*Proof.* We begin with the following identity (a special case of the polar identity for vector spaces):

$$(u - u^{k+1})^T M_k(u^k - u^{k+1}) = \frac{1}{2}(\|u - u^{k+1}\|^2_{M_k} - \|u - u^k\|^2_{M_k}) + \frac{1}{2}\|u^k - u^{k+1}\|^2_{M_k}.$$

We apply this to the VI formulation of the PDHG iteration (18) to get

$$h(u) - h(u^{k+1}) + (u - u^{k+1})^T Q(u^{k+1})$$
$$\geq \frac{1}{2}\left(\|u - u^{k+1}\|^2_{M_k} - \|u - u^k\|^2_{M_k}\right) + \frac{1}{2}\|u^k - u^{k+1}\|^2_{M_k}. \quad (24)$$

Note that

$$(u - u^{k+1})^T Q(u - u^{k+1}) = (x - x^{k+1})A^T(y - y^{k+1}) - (y - y^{k+1})A(x - x^{k+1}) = 0, \quad (25)$$

and so $(u - u^{k+1})^T Q(u) = (u - u^{k+1})^T Q(u^{k+1})$. Also, Assumption **C** guarantees that $\|u^k - u^{k+1}\|^2_{M_k} \geq 0$. These observations reduce (24) to

$$h(u) - h(u^{k+1}) + (u - u^{k+1})^T Q(u) \geq \frac{1}{2}\left(\|u - u^{k+1}\|^2_{M_k} - \|u - u^k\|^2_{M_k}\right). \quad (26)$$

We now sum (26) for $k = 0$ to $t - 1$, and invoke Lemma 3,

$$2\sum_{k=0}^{t-1}[h(u) - h(u^{k+1}) + (u - u^{k+1})^T Q(u)]$$

$$\geq \|u - u^t\|^2_{M_t} - \|u - u^0\|^2_{M_0} + \sum_{k=1}^{t}\left[\|u - u^k\|^2_{M_{k-1}} - \|u - u^k\|^2_{M_k}\right]$$

$$\geq \|u - u^t\|^2_{M_t} - \|u - u^0\|^2_{M_0} - 2C_\phi C_U - 2C_\phi C_H\|u - u^\star\|^2. \quad (27)$$

Because $h$ is convex,

$$\sum_{k=0}^{t-1} h(u^{k+1}) = \sum_{k=1}^{t} h(u^k) \geq th\left(\frac{1}{t}\sum_{k=1}^{t} u^k\right) = th(\tilde{u}_t).$$

The left side of (27) therefore satisfies

$$2t\left(h(u) - h(\tilde{u}_t) + (u - \tilde{u}_t)^T Q(u)\right) \geq 2\sum_{k=0}^{t-1}\left[h(u) - h(u^{k+1}) + (u - u^{k+1})^T Q(u)\right]. \quad (28)$$

Combining (27) and (28) yields the tasty bound

$$h(u) - h(\tilde{u}_t) + (u - \tilde{u}_t)^T Q(u) \geq \frac{\|u - u^t\|^2_{M_t} - \|u - u^0\|^2_{M_0} - 2C_\phi C_U - 2C_\phi C_H\|u - u^\star\|^2}{2t}.$$

Applying (19) proves the theorem. □

# 6  Numerical Results

We apply the original and adaptive PDHG to the test problems described in Section 3. We terminate the algorithms when both the primal and dual residual norms (i.e. $\|p^k\|$ and $\|d^k\|$) are smaller than 0.05. We consider four variants of PDHG. The method "Adapt:Backtrack" denotes adaptive PDHG with backtracking. The method "Adapt: $\tau\sigma = L$" refers to the adaptive method without backtracking with $\tau_0 = \sigma_0 = 0.95\rho(A^T A)^{-\frac{1}{2}}$.

We also consider the non-adaptive PDHG with two different stepsize choices. The method "Const: $\tau, \sigma = \sqrt{L}$" refers to the constant-stepsize method with both stepsize parameters equal to $\sqrt{L} = \rho(A^T A)^{-\frac{1}{2}}$. The method "Const: $\tau$-final" refers to the constant-stepsize method, where the stepsizes are chosen to be the final values of the stepsizes used by "Adapt: $\tau\sigma = L$." This final method is meant to demonstrate the performance of PDHG with a stepsize that is customized to the problem at hand, but still non-adaptive. The specifics of each test problem are described below:

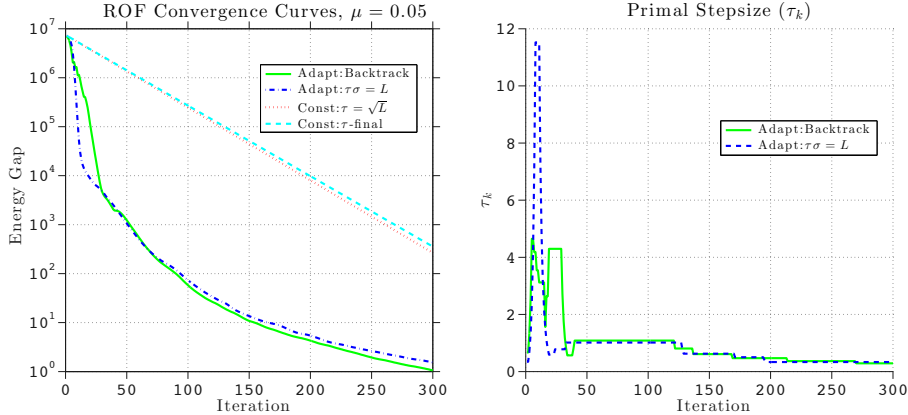

Figure 1: (left) Convergence curves for the TV denoising experiment with $\mu = 0.05$. The $y$-axis displays the difference between the objective (10) at the $k$th iterate and the optimal objective value. (right) Stepsize sequences, $\{\tau_k\}$, for both adaptive schemes.

Table 1: Iteration counts for each problem with runtime (sec) in parenthesis.

| Problem | Adapt:Backtrack | Adapt: $\tau\sigma = L$ | Const: $\tau, \sigma = \sqrt{L}$ | Const: $\tau$-final |
|---|---|---|---|---|
| Scaled Lasso (50%) | 212 (0.33) | 240 (0.38) | 342 (0.60) | 156 (0.27) |
| Scaled Lasso (20%) | 349 (0.22) | 330 (0.21) | 437 (0.25) | 197 (0.11) |
| Scaled Lasso (10%) | 360 (0.21) | 322 (0.18) | 527 (0.28) | 277 (0.15) |
| TV, $\mu = .25$ | 16 (0.0475) | 16 (0.041) | 78 (0.184) | 48 (0.121) |
| TV, $\mu = .05$ | 50 (0.122) | 51 (0.122) | 281 (0.669) | 97 (0.228) |
| TV, $\mu = .01$ | 109 (0.262) | 122 (0.288) | 927 (2.17) | 152 (0.369) |
| Compressive (20%) | 163 (4.08) | 168 (4.12) | 501 (12.54) | 246 (6.03) |
| Compressive (10%) | 244 (5.63) | 274 (6.21) | 908 (20.6) | 437 (9.94) |
| Compressive (5%) | 382 (9.54) | 438 (10.7) | 1505 (34.2) | 435 (9.95) |

**Scaled Lasso**   We test our methods on (8) using the synthetic problem suggested in [21]. The test problem recovers a 1000 dimensional vector with 10 nonzero components using a Gaussian matrix.

**Total Variation Minimization**   We apply the model (10) with $A = I$ to the "Cameraman" image. The image is scaled to the range $[0, 255]$, and noise contaminated with standard deviation 10. The image is denoised with $\mu = 0.25$, 0.05, and 0.01. See Table 1 for time trial results. Note the similar performance of Algorithm 1 with and without backtracking, indicating that there is no advantage to knowing the constant $L = \rho(A^T A)^{-1}$. We plot convergence curves and show the evolution of $\tau_k$ in Figure 1. Note that $\tau_k$ is large for the first several iterates and then decays over time.

**Compressed Sensing**   We reconstruct a Shepp-Logan phantom from sub-sampled Hadamard measurements. Data is generated by applying the Hadamard transform to a $256 \times 256$ discretization of the Shepp-Logan phantom, and then sampling 5%, 10%, and 20% of the coefficients are random.

# 7   Discussion and Conclusion

Several interesting observations can be made from the results in Table 1. First, both the backtracking ("Adapt: Backtrack") and non-backtracking ("Adapt: $\tau\sigma = L$") methods have similar performance on average for the imaging problems, with neither algorithm showing consistently better performance. Thus there is no cost to using backtracking instead of knowing the ideal stepsize $\rho(A^T A)$.

Finally, the method "Const: $\tau$-final" (using non-adaptive, "optimized" stepsizes) did not always outperform the constant, non-optimized stepsizes. This occurs because the true "best" stepsize choice depends on the active set of the problem and the structure of the remaining error and thus evolves over time. This is depicted in Figure 1, which shows the time dependence of $\tau_k$. This show that adaptive methods can achieve superior performance by evolving the stepsize over time.

# 8   Acknowledgments

This work was supported by the National Science Foundation ( #1535902), the Office of Naval Research (#N00014-15-1-2676), and the Hong Kong Research Grants Council's General Research Fund (HKBU 12300515). The second author was supported in part by the Program for New Century Excellent University Talents under Grant No. NCET-12-0111, and the Qing Lan Project.

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
