[Supplementary Material]

# Supplementary Material

## A    Proof of Lemma 1

*Proof.* Let

$$R(u) = \begin{pmatrix} \partial f(x) + A^T y \\ \partial g(y) - Ax \end{pmatrix}$$

be a vector containing the stacked primal and dual residuals (sub-gradients) for (1). Then the optimality condition for (1) can be written succinctly as

$$0 \in R(u^\star). \tag{29}$$

Using this notation, it can be seen that the iterates of PDHG satisfy

$$0 \in R(u^{k+1}) + M_k(u^{k+1} - u^k). \tag{30}$$

Subtracting (30) from (29) yields

$$M_k(u^{k+1} - u^k) \in R(u^\star) - R(u^{k+1}).$$

Now, $f$ and $g$ are convex and therefore $R$ is monotone. Taking the inner product with $(u^\star - u^{k+1})$ gives us

$$(u^\star - u^{k+1})^T M_k(u^{k+1} - u^k) \geq 0. \tag{31}$$

Now, observe the simple identity

$$\|u^k - u^\star\|_{M_k}^2 = \|u^{k+1} - u^k\|_{M_k}^2 + \|u^{k+1} - u^\star\|_{M_k}^2 + 2(u^k - u^{k+1})^T M_k(u^{k+1} - u^\star).$$

Applying (31) to this identity yields the result.    □

## B    Proof of Lemma 2

*Proof.* From Assumption **C**, we may assume without loss of generality that $Y$ is bounded (the case of bounded $X$ follows by nearly identical arguments). In this case, we have $\|y\| \leq C_y$ for all $y \in Y$.

Note that

$$\|u^{k+1} - u^\star\|_{M_{k+1}}^2 = -2(y^{k+1} - y^\star)^T A(x^{k+1} - x^\star) + \frac{1}{\tau_{k+1}} \|x^{k+1} - x^\star\|^2 + \frac{1}{\sigma_{k+1}} \|y^{k+1} - y^\star\|^2$$

$$\geq -2C_y\|A\|_{op}\|x^{k+1} - x^\star\| + \frac{1}{\tau_{k+1}} \|x^{k+1} - x^\star\|^2 + \frac{1}{\sigma_{k+1}} \|y^{k+1} - y^\star\|^2. \tag{32}$$

When $\|x^{k+1} - x^\star\|$ grows sufficiently large, the term involving the square of this norm dominates the value of (32). Since $\{\tau_k\}$ and $\{\sigma_k\}$ are bounded from above, it follows that there is some positive $C_x$ such that whenever

$$\frac{1}{\tau_{k+1}} \|x^{k+1} - x^\star\|^2 + \frac{1}{\sigma_{k+1}} \|y^{k+1} - y^\star\|^2 \geq C_x \tag{33}$$

we have

$$\frac{1}{\tau_{k+1}} \|x^{k+1} - x^\star\|^2 + \frac{1}{\sigma_{k+1}} \|y^{k+1} - y^\star\|^2 \geq 4C_y\|A\|_{op}\|x^{k+1} - x^\star\|. \tag{34}$$

Combining (34) with (32) yields

$$2\|u^{k+1} - u^\star\|_{M_{k+1}}^2 \geq \frac{1}{\tau_{k+1}} \|x^{k+1} - x^\star\|^2 + \frac{1}{\sigma_{k+1}} \|y^{k+1} - y^\star\|^2 \tag{35}$$

whenever (33) holds. In this case, we have

$$\|u^{k+1} - u^\star\|_{M_k}^2 = -2(y^{k+1} - y^k)^T A(x^{k+1} - x^\star) + \frac{1}{\tau_k} \|x^{k+1} - x^\star\|^2 + \frac{1}{\sigma_k} \|y^{k+1} - y^\star\|^2$$

$$\geq -2(y^{k+1} - y^\star)^T A(x^{k+1} - x^\star) + \frac{\delta_k}{\tau_{k+1}} \|x^{k+1} - x^\star\|^2 + \frac{\delta_k}{\sigma_{k+1}} \|y^{k+1} - y^\star\|^2$$

$$= \|u^{k+1} - u^\star\|_{M_{k+1}}^2 - \frac{\phi_k}{\tau_{k+1}} \|x^{k+1} - x^\star\|^2 - \frac{\phi_k}{\sigma_{k+1}} \|y^{k+1} - y^\star\|^2$$

$$\geq (1 - 2\phi_k)\|u^{k+1} - u^\star\|_{M_{k+1}}^2. \tag{36}$$

Applying (36) to Lemma 1, we see that

$$\|u^k - u^\star\|_{M_k}^2 \geq (1 - 2\phi_k)\|u^{k+1} - u^\star\|_{M_{k+1}}^2. \tag{37}$$

Note that $\lim_{k\to\infty} \phi_k = 0$, and so we may assume without loss of generality that $1 - 2\phi_k > 0$ (this assumption is only violated for finitely many $k$).

Now, consider the case that (33) does not hold. We have

$$\|u^{k+1} - u^\star\|_{M_k}^2 \geq \|u^{k+1} - u^\star\|_{M_{k+1}}^2 - \frac{\phi_k}{\tau_{k+1}}\|x^{k+1} - x^\star\|^2 - \frac{\phi_k}{\sigma_{k+1}}\|y^{k+1} - y^\star\|^2$$

$$\geq \|u^{k+1} - u^\star\|_{M_{k+1}}^2 - \phi_k C_x. \tag{38}$$

Applying (38) to Lemma 1 yields

$$\|u^k - u^\star\|_{M_k}^2 \geq \|u^{k+1} - u^\star\|_{M_{k+1}}^2 - \phi_k C_x. \tag{39}$$

From (37) and (39), it follows by induction that

$$\|u^0 - u^\star\|_{M_0}^2 \geq \prod_{i \in I_C} (1 - 2\phi_i)\|u^{k+1} - u^\star\|_{M_{k+1}}^2 - \sum_i \phi_i C_x \tag{40}$$

where $I_C = \{i \mid \frac{1}{\tau_{i+1}}\|x^{i+1} - x^\star\|^2 + \frac{1}{\sigma_{i+1}}\|y^{i+1} - y^\star\|^2 \geq C_x\}$. Note again that we have assumed without loss of generality that $i$ is large, and thus $1 - 2\phi_i > 0$.

We can rearrange (40) to obtain

$$\|u^{k+1} - u^\star\|_{M_{k+1}}^2 \leq \frac{\|u^0 - u^\star\|_{M_0}^2 + C_x \sum_i \phi_i}{\prod_i (1 - 2\phi_i)} < \infty$$

which shows that $\{\|u^k - u^\star\|_{M_k}^2\}$ remains bounded.

Finally, note that since $\{\tau_k\}$, $\{\sigma_k\}$, and $\{\|u^k - u^\star\|_{M_k}\}$ are bounded from above, it follows from (32) that $\{\frac{1}{\tau_k}\|x^k - x^\star\|^2\}$ is bounded from above. But $\{\frac{1}{\sigma_k}\|y^k - y^\star\|^2\}$ is also bounded from above, and so $\{\|u^k - u^\star\|_{H_k}^2\}$ is bounded as well. □