[Reviews · NeurIPS 2015]

Submitted by Assigned_Reviewer_1

This paper proposes a new adaptive primal-dual splitting method (PDHG) that can automatically tune the parameter values for convergence. My major concern is that existing literature, e.g., [6], has shown better results in terms of convergence/convergence rates.

Quality: This paper presents an interesting variant of PDHG that can automatically tune the parameter values for convergence. The technical part is mostly sound. However, some important details are missing. For example, this paper aims to solve (1), while the authors do not mention if the functions f(x) and g(y) are convex or not. According to the proof, I guess those functions may be assumed to be convex. Another example is the function h(Sx) in (6). The authors do not mention if h is convex or not. However, the first equation in page 3 (Fenchel conjugate of h) holds if and only if h is proper, closed, and convex.

Clarity: The authors may want to improve the readability. For example, a key ingredient of the proposed method is the backtracking condition in (14). The authors may want to briefly mention the intuitions and how it comes. Moreover, the authors may want to be more careful with notations. For the second equation in page 3, f(x) should be f(Ax-b) and y'Ax should be y'Sx. In Line 2 of Algorithm 1, do the authors really mean the norm of pk and dk?

Originality: I think the idea is novel when the authors first proposed this adaptive PDHG scheme.

Significance: My major concern is that existing literature has shown better results in terms of convergence/convergence rates. For example, [6] proposed a PDHG method with a convergence rate O(1/k^2), compared to the O(1/k) convergence rate of the method proposed in this paper. Therefore, to show the merit of this work, the authors may want to point out the advantage of the proposed adaptive PDHG to the method in [6]. Moreover, the experiments are a bit weak as the sizes of the data matrices are very small and the difference is only several seconds or milliseconds. The authors may want to evaluate their methods on several large-scale problems.
Summary: This paper proposes a new adaptive primal-dual splitting method (PDHG) that can automatically tune the parameter values for convergence. My major concern is that existing literature, e.g., [6], has shown better results in terms of convergence/convergence rates.

Submitted by Assigned_Reviewer_2

The problem addressed in this paper is the choice of the stepsize when using the alternating direction method of multipliers. The proposed solution is Adaptive PDHG (algorithm 1), a self-adaptive stepsize method for Primal-Dual Hybrid Gradient (PDHG), allowing reaching optimal convergence rate. The convergence of the proposed method is theoretically analyzed. Empirical evidence reported illustrates the interest of the proposed approach.

The paper is well-written, well-organized, self contained, easy to read and represents a significant piece of work. First part of the paper (sections 2 and 3) summarizes very clearly the Primal-Dual Hybrid Gradient Method and its application to sparse approximation in various machine learning contexts. Novelty, presented section 4 and 5, is the self-adaptive variant of Primal-Dual Hybrid Gradient method with theoretical guarantees for obtaining optimal convergence rate. Then empirical evidence reported could have been improved. The speeding advantage of the proposed approach may allow scaling over bigger problems. This issue is not included in the empirical evidence reported. More than the convergence speed, the significance of the proposed approach is the relief at not having to choose the stepsize.

Summary: The paper is well-written, well-organized, self contained, easy to read and represents a significant piece of work. Interesting and solid contribution may bit a bit marginal for the comunity.

Submitted by Assigned_Reviewer_3

This paper shows the convergence results of PDHG method with adaptive stepsize choices. An ergodic O(1/k) convergence rate is shown in the notion of variational inequality (Eq 19).

The presentation of the paper is clear, and the proof itself seems to give an answer to the questioned convergence of PDHG, which might be applicable to related settings: linearized ADMM and proximal point algorithms to solve saddle-point problems.

However, I think it needs to be clarified how well the suggest adaptive PDHG would perform compared to other algorithms for solving saddle point problems, e.g. Nesterov's smoothing or Tseng's accelerate proximal algorithm, which have similar O(1/k) ergodic convergence rate.

A minor note: in Eq (20), delta_{k-1} should be defined.
Summary: This paper provides a wanted convergence proof of adaptive PDHG, where the idea could be applied to other methods to solve saddle-point problems, which would be an important contribution to PDHG itself. However, the paper lacks of comparisons to other popular methods to solve SP problems and therefore it is not clear how appealing the method is against the competitors.

Submitted by Assigned_Reviewer_4

Augmented Lagrangian algorithms like ADMM have become popular recently since they decouple an objective function consisting of the sum of several functionals. PDHG is a version were the subproblems have a particularly simple form, more specifically, they are proximal problems w.r.t. the individual functionals of the objective function. In this paper, the authors propose an adaptive step selection rule for PDHG, provide convergence guarantees and show the numerical performance of the new algorithm.

The proposed algorithm seems to converge quickly according to the numerical experiments. However, there exist several PDHG algorithms which also use dynamic step sizes and have shown to outperform the methods used for comparison in the present paper. There is Algorithm 2 of [5] and the algorithm in "Adaptive Primal-Dual Hybrid Gradient Methods for Saddle-Point Problems" by T. Goldstein et al. (2013). The latter is particularly relevant here since it uses a very similar idea namely to balance the primal and dual residual. Without these comparisons it is a bit hard to judge how good the proposed algorithm actually is. A minor issue in this context is the question whether there are better constant step sizes than the two used in the experiments.

Another very relevant paper which is not cited here is "On the convergence of primal-dual hybrid gradient algorithms for total variation image restoration" by S. Bonettini and V. Ruggiero (2012) since it shows convergence of PDHG for dynamic step sizes.

Moreover, it would be useful to give a brief explanation of the result of Theorem 1. For example, one could clearly state that the nominator is bounded from above since u^k is bounded (Lemma 2). Also, providing the convergence result in terms of an average over {u^1,...,u^t} seems a bit unusual. Related to this is the question whether the sequences x^k or y^k converge.

Minor issues: * It should be \tilde{u}^T in Theorem 1. * C_\phi in Lemma 3 should be defined before it is used. * In (20), \delta_k-1 is not defined. One could directly replace (1-\delta_k-1) with \phi_k-1
Summary: This paper proposes a dynamic step size strategy for the PDHG algorithm, proves its convergence and shows advantages in convergence speed compared to several competing algorithms.

Author Feedback
Author rebuttal: Reviewer 1: We thank the reviewer for well thought-out comments. Fast convergence rates are certainly provable for non-adaptive schemes. However, the adaptive setting in much more challenging. For example, there are no known rates for general ADMM in the presence of adaptivity.
It is true that f and g are assumed convex. We absolutely agree that this should be explicit, and have already fixed this in our draft. Also, we thank the reviewer for catching or notation errors! We have fixed these.
Regarding originality, there has been some delay in publishing this work because we wanted to prove a rate (as opposed to simply convergence with no rate). Now that we have a rate, we are (finally) submitting this work for publication, and we think/hope the analysis is strong enough for a conference like NIPS!
We admire the work [6], and their results are strong for the non-adaptive case. However, our scheme has many advantages over non-adaptive methods, such as automated parameter tuning and a much less need for user oversight. Furthermore, our adaptive scheme could certainly be combined with an acceleration such as [6]. We will add text to the article elaborating on this.

Reviewer 2: We feel that a detailed comparison of PDHG to other schemes could easily be an entire paper, and is a bit out of the scope for a short theory paper. This is largely because PDHG does not solve the same class of problems as Nesterov's or Tseng's methods, and thus it is difficult to draw comparisons to PDHG without focusing on a very specific problem. Furthermore, there already exists some work comparing PDHG to other methods for specific problem classes (see Pock & Chambolle, ICCV '11). The purpose of our paper is not to explore when to use PDHG, but rather how to use adaptivity to simplify and speed up the implementation when PDHG is used.
That being said, we certainly take the reviewers comments very seriously, and agree that such a comparison is eventually needed. We are currently working on more extensive comparisons for a follow-on paper where more space is available.

Reviewer 3: We thank the reviewer for giving our paper a close read. In response to the comments, we feel that an in-depth comparison between PDHG and other schemes is probably beyond the scope of a short theory paper. Please see our comments to Reviewer 2.
Also, we do claim to be the first to introduce an adaptive PDHG variant. Algorithm 2 of [5] applies Nesterov-type acceleration to PDHG. However, the stepsize sequence in [5] is fixed a-priori, and is not adaptive - the user chooses the starting stepsizes, and all subsequent stepsizes are determined by this choice alone. With regards to the unpublished technical report "Adaptive Primal-Dual Hybrid Gradient Methods for Saddle-Point Problems," we ask the reviewer to see the third paragraph in our response to reviewer 1 ("Regarding...").
Finally, we will certainly add a reference to "S. Bonettini and V. Ruggiero" as this relates closely, and we have already made changes to the paper to clarify several issues that the reviewer (correctly) points out.

Reviewer 4: Comparisons to ADMM are not very relevant here because this paper mostly focuses on applications of PDHG where ADMM is not very effective. For example, consider the total variation problem (10) where A is a Hadamard matrix. ADMM would require expensive sub-steps to invert linear systems. Also, the scaled lasso problem has no "natural" ADMM formulation (although ADMM can be applied by using multiple "chained" splittings). In such situations ADMM has no hope of competing with PDHG.
That being said, there are certainly problems where ADMM is a very competitive (or even superior) choice to PDHG. An extensive comparison between algorithms is a bit out of scope (and length) for a short theory paper, although more extensive experiments will likely be included in an expanded follow-on paper.

Reviewer 5: We thank the reviewer for giving our paper a careful read. We think that PDHG is often a great choice for many problems. Consider the total variation problem (10) where A is a Hadamard matrix or DCT. ADMM would require expensive sub-steps to invert linear systems, whereas PDHG can solve every sub-problem is closed form. Also, the question of what algorithm is "best" is a balancing act between performance and practicality. Having a good adaptive variant of PDHG increases its practicality by making it more effective without user oversight.
The suggested citations are relevant, and we will fit them in the final version. Regarding the arxiv paper, we ask the reviewer to see the third paragraph in our response to reviewer 1 ("Regarding...").

Reviewer 6: We thank the reviewer for the well thought-out comments. The one weakness pointed out is that empirical results could be more extensive. We take the reviewer's opinions seriously, and will push for extensive results in a follow-on paper where more space is available.